# Etiologic Factors of Rotator Cuff Disease in Elderly: Modifiable Factors in Addition to Known Demographic Factors

**DOI:** 10.3390/ijerph19063715

**Published:** 2022-03-21

**Authors:** Ju Hyun Son, Zee Won Seo, Woosik Choi, Youn-Young Lee, Suk-Woong Kang, Chang-Hyung Lee

**Affiliations:** 1Department of Rehabilitation Medicine, Pusan National University Yangsan Hospital, Yangsan-si 50612, Korea; drsonjh85@gmail.com (J.H.S.); dr.adojiwon@gmail.com (Z.W.S.); dr.woosikchoi@gmail.com (W.C.); 2Woon-gok Liberal Arts Education College, Halla University, Wonju-si 26404, Korea; younyoung.lee@halla.ac.kr; 3Department of Orthopedics, Pusan National University Yangsan Hospital, Yangsan-si 50612, Korea; osksw98@gmail.com; 4Research Institute for Convergence of Biomedical Science and Technology, Pusan National University Yangsan Hospital, Yangsan-si 50612, Korea

**Keywords:** rotator cuff disease, etiology, isokinetic strength, osteoarthritis, elderly

## Abstract

With the aging society, musculoskeletal degenerative diseases are becoming a burden on society, and rotator cuff disease is one of these degenerative diseases. The purpose of this study was to examine the incidence of shoulder osteoarthritis and the etiologic factors of rotator cuff disease in the Korean elderly population. A total of 102 patients performing ultrasonography were recruited, and their demographic factors were analyzed. As functional factors, visual analog scale and the peak torque of external and internal rotators of the shoulder using an isokinetic dynamometer were measured. As an anatomical factor, the acromiohumeral distance in the plain radiograph of the glenohumeral anterior-posterior view was used. There were more female patients (65.7%) than male patients (34.3%). The age range with the highest number of respondents was 50–59 years old. The mean visual analogue score was 4.09 (Min 1 to Max 9). Age and dominant hand side factors appear to be the crucial etiologic factors of the presence and severity of rotator cuff disease. The lower net value of the external rotator strength is weakly related to the presence of rotator cuff disease after adjusting for age, and this is the only modifiable factor in the study.

## 1. Introduction

With the aging society, degenerative diseases of the musculoskeletal system are becoming more prevalent and a burden on society [1]. Osteoarthritis (OA) is the most prevalent degenerative disease, particularly among those older than 65, and one of the most common sources of pain and disability [2,3,4]. Numerous non-surgical options have been suggested for OA. Despite the many conservative treatments, unrelenting pain, loss of function, and advanced radiological changes have resulted in the need for surgical options [5,6,7,8,9]. In addition to these medical approaches, many therapeutic options, including health-related procedures, foods, and physical exercise, are used to manage degenerative diseases. In this sense, it is necessary to manage degenerative diseases from a national preventive level.

Unlike knee, hip, and spine joints, the shoulder joint is not influenced directly by the bodyweight. Rather, it is influenced by multifactorial reasons, including the functional use of the upper extremity in daily life [10]. Degenerative changes in the glenohumeral joints have been found in up to 17% of patients with shoulder pain, which has increased over the last 40 years [11,12]. The general prevalence of shoulder disease in the elderly is 5–17%, which is lower than that of other degenerative musculoskeletal diseases, including spine, hand, and knee (462 out of 696 (66%) in the spine, 415 out of 692 (60%) in the hand, 265 out of 696 (38%) in the knee, 36 out of 696 (5%) in the shoulder, and 15 out of 686 (2%), respectively) [13,14]. Because symptomatic glenohumeral osteoarthritis (GHOA) is lower than other weight-bearing musculoskeletal OA, the importance of the functional activity to reduce GHOA should be considered to reveal the precise relationship with shoulder OA.

The incidence of rotator cuff disease (RCD) is a strong etiologic factor causing shoulder OA [14]. RCD is a broad spectrum of pathology of rotator cuffs, including shoulder impingement syndrome, rotator cuff tendinopathy, partial to full thickness rotator cuff tear, calcification, and subacromial/subdeltoid (SASD) bursitis. The rotator cuff is made up of the muscle bellies and tendons of the infraspinatus, supraspinatus, teres minor, and subscapularis muscles. Symptoms are various according to pathologies and lesion sites, but most patients usually complain about pain with shoulder movement because of shoulder impingement [15]. Symptoms of RCD include pain with shoulder movement, especially when lifting arm overhead, night pain, pain with sleeping on lesion side, cracking sounds when moving arm, limitation in range of motion of shoulder, and rotator cuff muscle weakness [16,17]. In case of acute rotator cuff tears, patients complain of sudden weakness of the arm abduction. Diagnosis of RCD is supported by documentation supplied by radiologic studies, ultrasound examination, or magnetic resonance imaging examination [18]. A radiologic data study was related to the severity of RCD or OA [19,20,21]. The upward migration of the humeral head in radiology is associated with fatty degeneration of the supraspinatus and the tear size [22,23,24]. Tendon tears and fatty muscle degeneration in the rotator cuff correlate with a reduced acromiohumeral distance (AHD). The AHD at 90° of shoulder abduction was larger in asymptomatic patients with rotator cuff tear. In addition, tendon tears and fatty muscle degeneration in the rotator cuff correlate with reduced AHD. This anatomical change can also be an indirect sign of RCD development, which causes GHOA. Regardless of the severity of RCD, rotator cuff muscles could either be atrophic or undergo fatty degeneration and progress to RCD, which is a preceding condition to GHOA. In this sense, RCD is a crucial factor of shoulder degenerative disease, which is related directly to the incidence of shoulder OA. The relating factors for developing GHOA could be found by investigating the etiologic factors for developing RCD.

In addition to degenerative changes, muscle imbalance surrounding the rotator cuff has been suggested as a crucial cause of RCD [25,26,27]. Some studies reported an imbalance between the external rotator (ER) versus internal rotator (IR) strength in advanced RCD [27,28,29]. Thus, it is also important to reveal the relationship between muscle imbalance and the prevalence of GHOA. From previous reports, the development of RCD correlated with many relevant factors, including demographic, functional, and anatomical changes [18,30,31]. In addition, a dynamic imbalance in the shoulder muscle can affect the incidence of RCD; all influencing factors shoulder be considered in addition to age when evaluating the etiologic factors of GHOA. Theoretically, these various factors could either be etiologic or confounding factors according to age whenRCD develops. An examination of the etiology of RCD and the factors that can prevent it will help lower the prevalence of RCD in an aging society. We hypothesized that there are modifiable factors that affect developing RCD among functional or radiologic factors. Functional or radiologic factor can also influence the incidence of RCD greatly in addition to non-modifiable demographic factors such as age and dominant hand side.

Thus, the aim of this study was to investigate whether there are other modifiable etiologic factors of RCD in addition to currently known demographic factors.

## 2. Methods

### 2.1. Research Subjects and Data Collection Methods

One hundred and two unilateral RCD patients performing ultrasonography (US) using a 7.5 MHz linear transducer (LOGIQ E9; General Electric, Boston, MA, USA) in the authors’ clinic from October 2016 to May 2021 were recruited. The inclusion criteria were as follows: patients who have visited to our clinic from October 2016 to May 2021, adults above 18 years old, patients who was diagnosed with RCD by ultrasonography, and patientswho providedinformed written consent. The patients who complained of shoulder pain and weakness were examined by ultrasonography. After ultrasonography was taken, the presence of rotator cuff injury was included for the study.

An experienced physician, who is the specialist for physical medicine and has experienced 12,000 cases of shoulder ultrasonography per year for 21 years, measured the supraspinatus long- and short-axis using ultrasonography. The patient was in the modified Crass position, where the patient’s ipsilateral hand is placed on the closest hip or buttock region, and long-axis view of the supraspinatus by the transducer was obtained [32]. US image over superior facet of greater tuberosity shows hyperechoic and fibrillar supraspinatus tendon and the normal supraspinatus should be fibrillar and hyperechoic with a convex superior surface. Once the supraspinatus tendon is completely evaluated in the long axis, the transducer is turned 90° to obtain a short-axis view. Beginning with the transducer over the articular surface of the humeral head, the smooth, round echogenic surface of the humeral head and thin layer of hypoechoic hyaline cartilage with a uniform thickness of the overlying rotator cuff are seen.

The ultrasonic findings were as follows: RCD severity (0, I–II, III, and IV), presence of SASD bursitis, and presence of frozen shoulder. The RCD severity was defined as the supraspinatus tear severity in this study. Our ultrasonographic finding related to severity of the supraspinatus tendon tear was according to Snyder classification for RCD [33]. Supraspinatus tendon is “0” when normal cuff with smooth coverings of synovium and bursa. Severity of “I–II’ means minimal superficial bursal or synovial irritation or slight capsular fraying in a small, localized area which is usually lesser than 2 cm. Severity of “III” means more severe RCD, including fraying and fragmentation of tendon fibers, often involving the entire surface of a cuff tendon which is usually 2–3 cm. Severity of “IV” means very severe partial rotator tear that usually contains a sizeable flap tear in addition to fraying and fragmentation of tendon tissue, which is usually more than 4 cm.

The presence of bursitis was diagnosed when the thickness of subacromial/subdeltoid bursa more than 2 mm in supraspinatus long-axis view [34]. As there are no standard ultrasonographic criteria of frozen shoulder, the presence of frozen shoulder can be diagnosed clinically and radiologically. The inclusion criteria of frozen shoulder were: more than 50% decrease inrange of motion in flexion, external rotation, and internal rotation; presence of night pain or movement pain; presence of rotator cuff interval inflammation; and decrease inaxillary fold volume on ultrasonography.

As functional factors, pain in visual analog scale (VAS) and the peak torque of both ER and IRs of the shoulder were measured. Pain was divided into resting and movement pain using a VAS (range 0–10). The subjects were divided into three groups according to score: mild (VAS < 4), moderate (VAS 4–6), and severe (VAS > 6). An isokinetic dynamometer (System 4 Pro^TM^; Biodex Medical Systems, Inc., New York, NY, USA) was used to measure the peak torque of both the ER and IRs of the shoulder, respectively. When measuring isokinetic strengths of both ER and IRs of the shoulder, patients areon the firm table in the supine posture by fixation of the scapula and trunk. The tested arm wasabducted 90 degrees and elbow wasflexed 90 degrees. The lesion side ER and IR data were calculated as the ratio (ER/IR ratio).

As an anatomical factor, a radiologic specialist measured the AHD in the plain radiograph of the glenohumeral anterior-posterior (AP) view. One radiologic specialist reviewed all subjects’ plain radiograph of glenohumeral AP view and measured the AHD. We used the same distance measurement tool in picture archiving and communication system (PACS) by a radiologic specialist. In addition, to minimize the measurement bias, we set the same distance and angle during radiographs.

The exclusion criteria were as follows: patients who had previous shoulder surgery history, disease of shoulder instability, patients diagnosed with C5 or C6 radiculopathy or both, and lack of written informed consent of patient. This study was approved by the Pusan National University Yangsan Hospital Department of Health Institutional Review Board (IRB No. 05-2021-243). All the data were acquired from the patients by questionnaires and medical records after given informed consent.

### 2.2. Measurement of Variables and Statistical Analysis Method

Statistical analysis was performed using SPSS 18.0 (IBM, Armonk, NY, USA). The purpose of the study was to examine the factors related to the RCD. In order to achieve this purpose, statistical analysis was conducted according to the following procedure. First, frequency analysis and descriptive analysis were conducted to examine the demographic characteristics of the subjects in this study. Second, multiple correlation analysis was used to measure the strength of the linear relationship between each factor(i.e., RCD severity and sex, dominant hand or not, presence of frozen shoulder, calcific tendonitis, and SASD bursitis) and tried to explained the relationship between RCD and demographic factors. Third, a t-test was used for the comparison between two groups related to the RCD (i.e., sex, dominant hand side lesion, presence of frozen shoulder, presence of calcific tendonitis, presence of SASD bursitis). Finally, hierarchical regression analysis was conducted to determine the relationship between the RCD and several independent variables. In this hierarchical multiple regression analysis, the independent variable ‘age’ was entered first in the analysis of the data to investigate the relationship with the RCD. The second independent variable entered was dominant hand side and finally, SASD bursitis was put as the third independent variable to determine if this variable adds to the explanation of the RCD in addition to what is already explained by the prior independent variables.

## 3. Results

### 3.1. Subjects

According to an analysis of the demographic characteristics, there were more female patients (65.7%) than male patients (34.3%) (Table 1). The age range with the highest number of respondents was 50–59 years old, 60–69 years old, under 50 years old, and over 70 years old. The mean duration of pain in this study was 10.13 months (minimum 1 and maximum 120 months). The mean VAS was 4.09 (minimum 1 and maximum 9).

### 3.2. Analysis of Relationships between the Supraspinatus and Etiologic Factors

Table 2 lists the descriptive analysis of the independent variables.

Table 3 presents the results of the Pearson’s correlation analysis of the survey for the RCD severity and dominant hand frozen shoulder and calcific tendonitis of subjects that was conducted to identify the relationships among etiologic factors. The RCD severity and age showed a positive correlation with a Pearson’s correlation coefficient of r = 0.329 (*p* < 0.001), and other variables (BMI, VAS, strengths of both shoulder external and internal rotators, and AHD) showed no statistically significant correlations with each other.

As shown in Table 4, there was a significant difference in the RCD severity according to the presence or absence of the dominant hand and SASD bursitis, but there was no significant difference in sex, the presence of frozen shoulder, and calcific tendonitis.

### 3.3. Hierarchical Regression Analysis of the Supraspinatus and Etiological Factors

This study conducted the hierarchical regression analysis following the classification of the etiological factors in three dimensions to analyze the determinants of the RCD severity (Table 5). The determinants were identified through hierarchical regression analysis by inputting the explanatory variables across three stages. The demographic variable (i.e., sex, male = 0, Female = 1), dominant hand (if the dominant hand is the lesion side, the input was 0; if not correlated, the input was 1), and SASD bursitis (Absent = 0, Present = 1) variables were converted to dummy variables. Therefore, the changes in the etiological factors according to the supraspinatus were reviewed.

Hierarchical regression analysis was performed with three steps to investigate the effect on the RCD severity (Table 5). In the first step, the age variable was input (Model 1). In the second step, the presence of the dominant hand side lesion variable was also added (Model 2). Finally, in the third step, the presence of SASD bursitis was added for analysis. Model 1 (F = 12.154, *p* < 0.001), Model 2 (F = 11.829, *p* < 0.001), and Model 3 (F = 2.415, *p* < 0.01) all showed that the relationship between each variable and the RCD severity was significant. The level of explanatory power for each model also increased to 10.8% (R^2^ = 0.108) in Model 1, 19.3% (R^2^ = 0.193) in Model 2, and 21.2% (R^2^ = 0.212) in Model 3. In addition, an analysis of the tolerance (TOL) and variance inflation factor (VIF) to confirm the multicollinearity showed that the TOL was higher than 0.1. The variance inflation factors were below the threshold of 10, demonstrating no collinearity issues between the independent variables. Accordingly, in the case of Model 1, age (β = 0.329, *p* < 0.01) had a positive (+) influence on the RCD severity, meaning that there was a significant correlation between age and the RCD severity. In Model 2, age (β = 0.329, *p* < 0.01) had a positive relationship with the RCD severity and a negative relationship with the presence of the dominant hand (β = −0.294, *p* < 0.01). These results were significant. Finally, in the results of Model 3, in which the presence of SASD bursitis variable was added, the severity of RCD increased significantly with age and the lesion on the dominant-hand side. On the other hand, there was no significant relationship between the presence of SASD bursitis and RCD severity.

The peak torque isokinetic strength of ER and IR at both the sound and lesion sides were compared between groups of presence and absence of RCD to examine the relationship between the presence of RCD and shoulder muscle imbalance (Table 6). Only the ER strength of the RCD group was significantly lower than that of the no RCD group (t = 2.232, *p* = 0.028). Regression analysis was performed between the ER strength and the presence of RCD after adjusting the age factor. With adjusting for age, the lesion-side ER strength in the RCD group decreased significantly (B = −0.085, *p* = 0.039) (Table 7).

## 4. Discussion

From previous reports, musculoskeletal OA also increases with age [35,36,37,38]. Nevertheless, the prevalence of shoulder OA is lower than that of other musculoskeletal diseases that burden axial loading. Thus, we hypothesized that the prevalence of shoulder OA increases with age to some extent, and other factors could also influence the development of OA. Among many etiologic factors, RCD has been known to be one of the strong factors of OA [14]. As the shoulder joint is a ball and socket joint and is surrounded by rotator cuff muscles, the muscular dynamic balance could also influence the degeneration of RCD. We thought that the development of RCD might bethe sum of modifiable and non-modifiable factor interactions. In this sense, we put the biomechanical factor, the net and ratio of rotator cuff strength, into etiologic factors and compared it with other primary etiologic factors, such as radiologic factor, age, and dominant hand. In this sense, this study focused the muscle imbalance in addition to other well-known etiologic factors to cause RCD in the Korean population along with age.

### 4.1. Demographic Factors

A previous study reported the RCD-related demographic factors, such as age, sex, and dominant arm [18,30,31]. In concordance with many previous reports on the demographics related to the development of RCD in previous studies, this study did not suggest strong relating factors of RCD, except for age. One study showed that the prevalence of rotator cuff tears in Japan was 20.7%, and the prevalence increased with age, dominant arm, and a history of trauma [39]. A recent systemic review suggested the hand dominance and older age are significantly associated with rotator cuff tears [40]. Age above 60 years, dominant hand, overweight, smoking, and hypertension could also be etiologic factors of RCD [40]. They reported that factors such as diabetes mellitus, hypertension, smoking, and older age prompt such chemical signals that could induce excessive apoptosis and impede current elimination pathways via poor vascularity, alterations in material properties, and matrix composition changes [40]. One study examined the prevalence of shoulder OA in Korea [41]. They reported that approximately 16.1% of the elderly (aged 75 years and older) had primary OA confirmed by radiographs. On the other hand, it only showed a high incidence of OA in the elderly and did not compare the gradual incidence according to age.

### 4.2. Functional Factors: Rotator Cuff Muscle Imbalance

According to the original assumption that functional impairment could also be an important factor causing RCD in addition to age, this study investigated the muscle imbalance of the rotator cuff muscles. In previous reports, the related functional factors, such as muscle imbalance of the rotator cuff muscles, mainly in the horizontal plane, were reported to cause RCD [42,43]. The imbalance between ER/IR could yield RCD [44]. Therefore, ER/IR imbalance could be regarded as a crucial factor of RCD. Repetitive overhead throwing, which was performed in sports, may result in muscle imbalance between the ER and IRs [44]. The fatty degeneration of the rotator cuff muscle correlates with the isokinetic muscle strength deficit, but the tear extent was not reflected directly to it [45].

Sometimes, the muscles of the shoulder may become imbalanced by injury or atrophy, which can cause the shoulder to move forward with certain activities and again cause impingement [46]. With aging, this muscle imbalance can worsen, leading to tendon wear, irreversible fatty infiltration of the rotator cuff muscles, and upward migration of the humeral head [47]. Muscle fatigue can develop in RCD [47,48]. In particular, deltoid muscle fatigue is believed to be related to the weakness of the supraspinatus and subscapularis [47]. In its natural state, the shoulder is unbalanced in both the vertical and horizontal planes because the deltoid is stronger than the rotator cuff muscles, and the internal rotator muscles are stronger than the ER muscles. From the previous studies, one can think that RCD has a degenerative tendency; however, this tendency isnot precisely correlated with age. Although this study did not show a significant association between the RCD severity and ER/IR ratio (Table 3). On the other hand, the net ER strength of the lesion side showed a positive correlation with RCD in regression analysis when adjusting the age factor.

### 4.3. Radiographic Factors

Although RCD can decrease the AHD, the AHD of the symptomatic shoulder in recreational athletes is not significantly decreased [49]. A decrease in the AHD in symptomatic shoulders is not significantly associated with increased pain or functional limitations in recreational athletes [49]. If overhead activity occurs repetitively, especially in athletes, one can assume that RCD could be developed [50]. In addition, the simple presence of hooked or curved acromion could not be a crucial factor of RCD development [51]. In this sense, the radiology findings were not associated directly with the severity of the RCD, as the pathology is not severe in the general population.

The clinical importance of this study is that the multifactorial causes of RCD were examined in a Korean population. In addition to the aging factor as a cause of RCD, demographic, functional, and radiological factor analyses were used to reveal the precise relationship of RCD development. In particular, these factors need to be considered in an aging society. These findings also showed that age and dominant hand factors were the only statistically significant factors of RCD development, thus eventually correlated with OA. Although there was a weak correlation, the net strength of ER should also be considered in preventing the development of RCD in the elderly.

This study had several limitations. First, only the supraspinatus tendon pathology was regarded as the RCD severity. Severe RCD pathologies, including more than two muscle diseases, were excluded to allow for a clear investigation. In addition, previous surgical operation or unstable patients were also excluded. Thus, the actual prevalence of the rotator cuff pathology from various etiologies would be greater than assumed. Second, in the radiology findings, the AHD and supraspinatus tendon tear thickness were measured using X-ray and US, respectively. The morphological size of the distance and tendon tear may not be precisely related to the functional muscle weakness to cause RCD. Third, other systemic factors, including hypertension, diabetes, and thyroid function, were not investigated. Although these systemic factors are not crucial factors to cause RCD, further study will be needed to determine the relationship between all the demographic factors and RCD development.

## 5. Conclusions

Age and dominant hand side factors appear to be the crucial etiologic factors of the presence and severity of RCD. The lower net value of the ER strength is weakly related to the presence of RCD after adjusting for age. Despite the degenerating process with aging, consideration of various factors should be needed to retard the RCD development in an aging society.

## Figures and Tables

**Table 1 ijerph-19-03715-t001:** Characteristics and descriptive analysis of the subjects (*n* = 102).

Variable	Category	Subjects (%)	Mean ± SD
Sex	Male	35 (34.3)	-
	Female	67 (65.7)	-
Age (years)	Under 50	16 (15.7)	-
	50–59	40 (39.2)	-
	60–69	35 (34.3)	-
	Over 70	11 (10.8)	-
Body weight (kg)	40–49	12 (11.8)	-
	50–59	38 (37.3)	-
	60–69	33 (32.4)	-
	70–79	13 (12.7)	-
	80–89	6 (5.9)	-
Height (cm)	Under 150	8 (7.8)	-
	150–159	38 (37.3)	-
	160–169	47 (46.1)	-
	Over 170	9 (8.8)	-
VAS	Mild	24 (23.8)	-
	Moderate	74 (73.3)	-
	Severe	3 (3.0)	-
Duration of pain (months)		-	10.13 ± 17.03
VAS		-	4.09 ± 1.05
ER of sound side shoulder (N·m)		-	12.16 ± 6.70
ER of lesion side shoulder (N·m)		-	8.90 ± 6.04
IR of sound side shoulder (N·m)		-	18.07 ± 8.12
IR of lesion side shoulder (N·m)		-	15.57 ± 8.15
ER/IR ratio of lesion side shoulder		-	0.58 ± 0.33
AHD (mm)		-	9.41 ± 2.03

VAS: visual analog scale; Mild: VAS 1–3; Moderate: VAS 4–6; Severe: VAS 7–10;ER: isokinetic strength peak torque of shoulder external rotator in 60°/s; IR: isokinetic strength peak torque of shoulder internal rotator in 60°/s; AHD: acromiohumeral distance; SD: standard deviation.

**Table 2 ijerph-19-03715-t002:** Descriptive analysis of each factor confirmed by ultrasonography (*n* = 102).

Factors		Subjects	Proportion (%)
Lesion side	Left	35	34.3
	Right	67	65.7
Dominant	Dominant hand	32	31.4
	Non-dominant hand	70	68.6
Frozen shoulder	Absent	15	14.7
	Present	87	85.3
Calcific tendonitis	Absent	91	89.2
	Present	11	10.8
Subacromio/subdeltoid bursitis	Absent	72	70.6
	Present	30	29.4
RCD severity	0	26	25.5
	I–II	43	42.2
	III	23	22.5
	IV	10	9.8

RCD: rotator cuff disease; RCD severity means supraspinatus tendon tear according to Snyder’s classification.

**Table 3 ijerph-19-03715-t003:** Pearson’s correlation coefficients between RCD and each factor (*n* = 102).

	Age	VAS	Sound ER	Lesion ER	Sound IR	Lesion IR	Lesion ER/IR Ratio	AHD
RCD severity	0.329 ***	0.056	−0.130	−0.105	−0.023	−0.052	−0.022	−0.112

RCD: rotator cuff disease; VAS: visual analog scale; Sound ER: isokinetic strength peak torque of unaffected shoulder external rotator in 60°/s; Lesion ER: isokinetic strength peak torque of affected shoulder external rotator in 60°/s; Sound IR: isokinetic strength peak torque of unaffected shoulder internal rotator in 60°/s; Lesion IR: isokinetic strength peak torque of affected shoulder internal rotator in 60°/s; Lesion ER/IR ratio: the ratio of Lesion ER to Lesion IR; AHD: acromiohumeral distance; *** *p* < 0.001.

**Table 4 ijerph-19-03715-t004:** Difference analysis according to the relevant factors of the RCD severity.

Dependent Factor	Independent Factor	Group	*N*	Mean	SD	T Value
RCD severity	Gender	Male	35	1.23	0.973	0.487
		Female	67	1.13	0.903	
	Dominant hand side lesion	Dominant	32	1.63	0.976	3.363 **
		Non-dominant	70	0.96	0.824	
	Presence of frozen shoulder	Absent	15	0.93	0.799	−1.060
		Present	87	1.21	0.942	
	Presence of calcific tendonitis	Absent	91	1.18	0.950	0.287
		Present	11	1.09	0.701	
	Presence of SASD bursitis	Absent	72	1.28	0.982	2.171 *
		Present	30	0.90	0.712	

RCD: rotator cuff disease; SASD: subacromio/subdeltoid; SD: standard deviation; * *p* < 0.05, ** *p* < 0.01.

**Table 5 ijerph-19-03715-t005:** Hierarchical regression analysis.

**Model 1**	**B**	**SE**	**β**	**T(p)**
(Constant)	−0.662	0.532		−1.245
Age	0.032	0.009	0.329	3.486 **
F(p)	12.154 **			
Adj. R^2^	0.099			
△R^2^	0.108			
**Model 2**	**B**	**SE**	**β**	**T(p)**
(Constant)	−0.015	0.547		−0.028
Age	0.027	0.009	0.285	3.117 **
Dominant hand side	−0.583	0.181	−0.294	−3.220 **
F(p)	11.829 ***			
Adj. R^2^	0.177			
△R^2^	0.085			
**Model 3**	**B**	**SE**	**β**	**T(p)**
(Constant)	0.079	0.546		0.145
Age	0.027	0.009	0.279	3.073 **
Dominant hand side	−0.553	0.181	−0.279	−3.062 **
SASD bursitis	−0.283	0.182	0.14	−1.554
F(p)	8.804 ***			
Adj. R^2^	0.188			
△R^2^	0.019			

SASD: subacromio/subdeltoid; ** *p* < 0.01, *** *p* < 0.001.

**Table 6 ijerph-19-03715-t006:** T-test of isokinetic strengthening of the ER and IR at both sound and lesion sides and presence of RCD.

		N	Mean	SD	t(p)	*p*-Value	Mean Diff	SD Diff	95% Confidence Interval	
									Lower Limit	Upper Limit
Lesion ER	Absence of RCD	26	11.146	7.29	2.232 *	0.028	3	1.34	0.334	5.681
	Presence of RCD	76	8.13	8.25						
Sound ER	Absence of RCD	26	13.623	7.16	1.29	0.2	1.95	1.51	−1.053	4.973
	Presence of RCD	76	11.663	6.51						
Lesion IR	Absence of RCD	26	17.431	6.66	1.351	0.18	2.49	1.84	−1.168	6.156
	Presence of RCD	76	14.937	8.55						
Sound IR	Absence of RCD	26	18.292	8.25	0.155	0.877	0.28	1.85	−3.391	3.966
	Presence of RCD	76	18.005	8.13						

RCD: rotator cuff disease; Lesion ER: isokinetic strength peak torque of affected shoulder external rotator in 60°/s; Sound ER: isokinetic strength peak torque of unaffected shoulder external rotator in 60°/s; Lesion IR: isokinetic strength peak torque of affected shoulder internal rotator in 60°/s; Sound IR: isokinetic strength peak torque of unaffected shoulder internal rotator in 60°/s; SD: standard deviation; diff: difference; * *p*< 0.05.

**Table 7 ijerph-19-03715-t007:** Regression analysis between the presence of RCD and ER strength by adjusting for age.

	B	S.E.	Wald	Degree of Freedom	*p*-Value	Exp(B)
ER	−0.085	0.041	4.251	1	0.039 *	0.918
Age	0.069	0.028	6.262	1	0.012 *	1.072
Constant	−2.057	1.575	1.705	1	0.192	0.128

RCD: rotator cuff disease; ER: isokinetic strength peak torque of affected shoulder external rotator in 60°/s; * *p* < 0.05.

## Data Availability

The data presented in this study are available on request from the corresponding author. The data are not publicly available due to patients’ privacy.

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
