# Peer review of "Etiologic Factors of Rotator Cuff Disease in Elderly: Modifiable Factors in Addition to Known Demographic Factors"

_ijerph, 2022, doi:10.3390/ijerph19063715_

Round 1
Reviewer 1 Report
Dears Authors,
in relation to the previous version, the article has changed.
General information:
-Tittle: OK
Introduction:
-rotator cuff disease (RCD) should be explained: symptoms, diagnosis criteria, characteristics, etc. lack this information, it should be briefly described
-References are newer, OK.
-In text which you added: "Diagnosis of a RCD is based upon the evaluation of appropriate symptomsclinical features of shoulder pain and limitation of range of motion during overhead activitiesand physical findings, supported by documentation supplied by radiologic studies, ultrasound examination, or magnetic resonance imaging examination[18]"
What kind of symptoms ? There is no need to describe. Just replace them.
-There is no hypothesis of study: The hypothesis is not clear still, it doesn't correlate with the title of the article. It should be corrected.
Aim of study: The aim of the research is not clear to me, it doesn't correlate with the title of the article. It should be corrected.
Possibly the title of the article may need to be corrected.
M&M:
-modified Crass position - please briefly present it, description of the position or photo, it may be unclear to the reader
-Inclusion criteria. Please present in the text more clearly, e.g. in points...
Patient give their written consent to the study, so please give the information in Inclusion criteria (written consent of the patient) or in exclusion criteria (lack of..)
-The M&M are described too generally, many elements of the study should be specified.. how the measurement was performed, in which units the measurement was performed, what are the criteria of physiology and pathology, etc: OK
-Why is the age limit of 50 years old..? OK
Result:
3.1. Subjects is a mistake: female (67.5%) than male (34.3%) / total is 101,8 % - have to corrected it. OK
Information contained in the tables should not be repeated in the text. OK
In table 1: Subjects (%) or Mean ± SD / please make it clearer - which is subjects / which is Mean - the reader may not understand this. OK
"The mean duration of pain in this study was 10.13 ± 9.64 years" - please describe it in more detail, give the minimum and maximum value.
"The mean VAS was 4.09±1.05" - please describe it in more detail, give the minimum and maximum value. I think that it can be important informations.
Legend in table 1 the same information is repeated. OK
In table 1 the units should be explained. OK
In table 2 please explain RCD in legend. OK
In table 2: How "Tendinopathy", "Partial thickness tear" is different from "Full thickness tear" - please explain/ describe it. Please use the number to describe Snyder classification: I -... II - ... III-... IV - ... Then it will be clear to the reader. OK
In table 3: all abbreviations should be explained. OK
In table 3,4,5,6 7: the presented numerical values are incorrect / unclear. Should be: 0.12345... etc. OK
The legend in table 3 should be completed. It could be present clearer.
The legend in table 4,5,6 and 7 should be completed. OK
In table 4, 5,6 and 7: all abbreviations should be explained. OK
Discussion:
References should be in the order of citation / first 26-28 after 29-32. OK
The discussion needs to be supplemented / expanded / newer citations should be added after 2015 / the presented discussion is insufficient.
The references are newer. Presented discussion is better. OK
Author Response
Introduction:
-rotator cuff disease (RCD) should be explained: symptoms, diagnosis criteria, characteristics, etc. lack this information, it should be briefly described
-> We added a sentence of explanation for RCD briefly. But RCD is a broad spectrum of pathology, therefore there is lack of consensus on clinical assessment of RCD yet.
“Rotator cuff disease is a broad spectrum of pathology of rotator cuffs, including shoulder impingement syndrome, rotator cuff tendinopathy, partial to full thickness rotator cuff tear, calcification and subdeltoid-subacromial bursitis. The rotator cuff is made up of the muscle bellies and tendons of the infraspinatus, supraspinatus, teres minor, and subscapularis muscles. Symptoms are various according to pathologies and lesion sites, but most patients usually complain pain with shoulder movement because of shoulder impingement. “
-In text which you added: "Diagnosis of a RCD is based upon the evaluation of appropriate symptoms clinical features of shoulder pain and limitation of range of motion during overhead activities and physical findings, supported by documentation supplied by radiologic studies, ultrasound examination, or magnetic resonance imaging examination[18]"
What kind of symptoms ? There is no need to describe. Just replace them.
-> We added relating sentence and corrected written sentences.
“Symptoms of RCD include pain with shoulder movement, especially when lifting arm overhead, night pain, pain with sleeping on lesion side, cracking sounds when moving arm, limitation in range of motion of shoulder and rotator cuff muscle weakness.”
-There is no hypothesis of study: The hypothesis is not clear still, it doesn't correlate with the title of the article. It should be corrected.
Aim of study: The aim of the research is not clear to me, it doesn't correlate with the title of the article. It should be corrected.
Possibly the title of the article may need to be corrected.
-> We corrected whole title, hypothesis, and aim of study.
Hypothesis: We hypothesized that there are modifiable factors that affect developing RCD among functional or radiologic factors. Functional or radiologic factor, also could influences the incidence of RCD greatly in addition to non-modifiable demographic factors such as age and dominant hand side.
Aim of study: Thus, the aim of this study was to investigate whether there are other modifiable etiologic factors of RCD in addition to currently known demographic factors.
M&M:
-modified Crass position - please briefly present it, description of the position or photo, it may be unclear to the reader
-> We added sentence of describe it.
“where the patient’s ipsilateral hand is placed on the closest hip or buttock region”
-Inclusion criteria. Please present in the text more clearly, e.g. in points...
Patient give their written consent to the study, so please give the information in Inclusion criteria (written consent of the patient) or in exclusion criteria (lack of..)
à We added the criteria in both inclusion and exclusion criteria clearly.
“The inclusion criteria were as follows; patients who have visited to our clinic from October 2016 to May 2021, adults above 18 years old, patients who was diagnosed with RCD by ultrasonography and subjects who received informed written consent of patients.”
“The exclusion criteria were as follows; patients who had previous shoulder surgery history, disease of shoulder instability, patients diagnosed with C5 or C6 radiculopathy or both, and lack of written informed consent of patient.”
Reviewer 2 Report
The authors provide sufficient answers to my comment.
Author Response
Thank you for your comment.
Reviewer 3 Report
Thank you for submitting the revised manuscript.
I believe that the manuscript has been well revised and is now more understandable to readers.
I have no additional comments.
The long-term effects of strength-balance interventions on RCD development, the only modifiable factor for RCD development shown in this study, are of great interest and I look forward to continued research.
Author Response
Thank you for your comment.